SNARC-like effect for tempo is consistent for fast and full tempo ranges but still controversial for slow tempo range

Mariconda Alberto alberto.mariconda@phd.units.it 1
Murgia Mauro 1
De Tommaso Matteo 2
Agostini Tiziano 1
Prpic Valter valter.prpic@unibo.it 3 4
1 Department of Life Sciences, University of Trieste , Trieste , Italy
2 Department of Psychology, University of Milan, Bicocca , Milano , Italy
3 Department of Philosophy, University of Bologna , Bologna , Italy
4 Institute for Psychological Sciences, De Montfort University , Leicester , United Kingdom
Dalmaso Mario
Electronic publication date: 2024 Sep 18
Publication date: 2024
Volume: 12
Electronic Location ID: e18009
Received 2024 Apr 30; Accepted 2024 Aug 9
Copyright: ©2024 Mariconda et al.
Copyright year: 2024
Copyright holder: Mariconda et al.
License: This is an open access article distributed under the terms of the Creative Commons Attribution License, which permits unrestricted use, distribution, reproduction and adaptation in any medium and for any purpose provided that it is properly attributed. For attribution, the original author(s), title, publication source (PeerJ) and either DOI or URL of the article must be cited.
License URL: https://creativecommons.org/licenses/by/4.0/

Keywords: Time, Music tempo, Mental timeline, Bpm, Music, Full tempo range, Slow tempo range, Fast tempo range, SNARC effect, SNARC-like effect

Funding: The authors received no funding for this work.

==============================
Recent evidence suggested the existence of a spatial associations for music tempo with faster left-hand responses to relatively slow tempos and faster right-hand responses to relatively fast tempos. We refer to a study that systematically explored these spatial associations across different tempo ranges, revealed a clear effect only in the fast tempo range (DOI 10.3758/s13414-019-01945-8). The present study further investigated whether a spatial association exists across different tempo ranges (i.e., “full”, “slow” or “fast” tempo range). In particular, the present study was conducted aiming (1) to test the spatial associations for tempo in the full tempo range (Experiment 1) and (2) to further investigate the occurrence of this spatial associations in the slow and fast tempo ranges (Experiment 2). Experiment 1 revealed a spatial association for tempo occurs in the full tempo range (40–200 bpm). Experiment 2 confirmed this association in the fast tempo range (133–201 bpm) but showed contradictory results in the slow tempo range (40–104 bpm). This suggests that a spatial association is plausible in the slow tempo range, although further research is needed to clarify this phenomenon.

Introduction

Increasing evidence in cognitive psychology suggests that people tend to mentally represent time as a spatial continuum on a line oriented from left to right (the mental timeline or “MTL”; Bonato, Zorzi & Umiltà, 2012). This representation emerges when time is conveyed both semantically by words and verbs (e.g., “yesterday” vs. “tomorrow” or “he said” vs. “he will say”; Santiago et al., 2007; Torralbo, Santiago & Lupiáñez, 2006) and by perceptual stimuli (i.e., non-verbal-stimuli for which is possible the manipulation of temporal parameters, such as time duration Ishihara et al., 2008; Mariconda et al., 2022). Evidence suggests that the spatial alignment of this mental timeline is influenced by embodied and cultural factors, which are closely linked (Casasanto, 2016). In the absence of specific experimental manipulations, the direction of the MTL typically follows the writing/reading direction. Specifically, the left-to-right writing/reading direction reflects the physical movement of our hands/eyes but also aligns with the temporal flow from left (past events/short durations) to right (future events/long duration). This tendency can be altered by providing instructions in a right-to-left format for individuals who typically read from left to right (Casasanto & Bottini, 2010).

This interplay between time and space is commonly revealed by response time advantages in experimental tasks. For instance, evidence with verbal stimuli showed a left response time advantage for past-related words and right response time advantage for future-related words (Torralbo, Santiago & Lupiáñez, 2006; Ulrich & Maienborn, 2010; Weger & Pratt, 2008; for an overview see Eikmeier et al., 2016). Similarly, evidence with perceptual stimuli revealed left (vs. right) response hand advantages for short (vs. long) durations (Vallesi, Binns & Shallice, 2008). Similar spatial associations have been identified also along the vertical and diagonal axes (Dalmaso, Schnapper & Vicovaro, 2022; Topić, Stojić & Domijan, 2022). Additionally, some evidence showed a left (vs. right) response hand advantage for early (vs. late) temporal stimuli (i.e., stimuli presented either earlier or later than expected occurrence within a periodic sequence of auditory stimuli; Ishihara et al., 2008; Mariconda et al., 2022).

The spatial associations for time can be explained within the theoretical framework of the ATOM (“A Theory of Magnitude”) model proposed by Walsh (Walsh, 2003; Bueti & Walsh, 2009). This model suggests that space, time and quantities are processed by a shared magnitude system, causing response biases between these dimensions. The most known of these effects is the Spatial-Numerical Association of Response Codes (SNARC) effect (Dehaene, Bossini & Giraux, 1993), which demonstrates that digits are spatially represented in the human mind (for alternative accounts see Proctor & Cho, 2006; Van Dijck & Fias, 2011). Follow up studies extended this phenomenon also to non-symbolic numerosities (Cutini et al., 2019; Nemeh et al., 2018; Nuerk, Wood & Willmes, 2005; Prpic et al., 2023) and other non-numerical magnitudes (for an overview, see Prpic et al., 2021). For instance, in the visual modality, SNARC-like effects have been observed for luminance, size of pictorial figures, and facial expressions of emotions (Baldassi et al., 2021; Fantoni et al., 2019; Fumarola et al., 2014; Holmes & Lourenco, 2011; Prpic et al., 2020; Ren et al., 2011). All these effects fell under the prediction of ATOM which posits that any kind of quantity, independently from its format, should be spatially coded and elicit a Spatial Quantity Association of Response Codes (SQUARC) effect.

Particularly relevant for the present work are studies in the music context, which demonstrated that different features of sound can be mapped into space. Although few studies looked into SNARC-like effects for music notation (Ariga & Saito, 2019; Fumarola et al., 2020; Prpic et al., 2016), most of the work in the field focused on the auditory properties of sound, such as pitch height (Guida & Porret, 2022; Lega et al., 2020; Lidji et al., 2007; Pitteri et al., 2017; Prpic & Domijan, 2018; Rusconi et al., 2006), loudness (Bruzzi et al., 2017; Hartmann & Mast, 2017) and tempo (De Tommaso & Prpic, 2020; Mariconda et al., 2024). In the present work we specifically focused on tempo.

In music context, the term “tempo” is defined as the speed of an auditory sequence and is typically expressed in beats per minute (bpm). Since tempo is a fundamental element in music, De Tommaso & Prpic (2020) investigated whether tempo could elicit a SNARC-like effect when participants were asked to determine if a target sequence of beats was faster or slower than a reference. The authors conducted three experiments aiming to systematically explore a full tempo range (40, 80, 160, 200 bpm; Experiment 1), a slow tempo range (40, 56, 88, 104 bpm; Experiment 2) and a fast tempo range (133, 150, 184, 201 bpm; Experiment 3). Surprisingly, the results revealed a SNARC-like effect only with in the fast tempo range (Experiment 3). These results suggest that, like other continuous quantities, tempo is spatially coded. However, differently from other spatial continua, tempo might be limited to a specific range of stimuli suggesting that slow and fast tempos could be represented differently through space.

In a follow up commentary, Wood, Shaki & Fischer (2021) raised some concerns regarding the work by De Tommaso & Prpic (2020). Specifically, Wood and colleagues pointed out that participants could complete the task even before being exposed to the second or third beat of the sequence. This was particularly evident when evaluating the slowest tempos, thus suggesting the occurrence of anticipatory responding. Consequently, the anticipatory responding may have led participants to focus on the interval duration between beats rather than judging the temporal speed, thus creating a conflict and potentially contributing to the absence of spatial associations with slower tempos. However, Mariconda et al. (2024), by using a novel paradigm with only two beats in sequence, addressed the issue of anticipatory responding. In that study participants were either required to judge the temporal speed of the sequences (as “slow” or “fast”) or the interval duration between the two beats (as “short” or “long”) in separate conditions. The results demonstrated that the SNARC-like effect for tempo is driven by temporal speed even when participants explicitly focus on interval duration. Nonetheless a limitation of the study by Mariconda et al. (2024) is that their paradigm only provides evidence regarding fast tempos, precluding direct inferences about slow tempos.

The aim of the present study is twofold. First, to extend the study by De Tommaso & Prpic (2020) to test the robustness of the SNARC-like effect for tempo. Second, to address some of the issues of previous studies and extend the investigation to slow tempos. Specifically, in Experiment 1, we extended Experiment 1 by De Tommaso & Prpic (2020) using a higher number of stimuli within the same range of bpm, to enhance a continuous representation of the temporal sequence. In Experiment 2, we investigated the occurrence of SNARC-like effects when separately testing the stimuli in the slow and fast tempo ranges, using a within-participants design. This would allow to control for the individual variability which might have led to different results in previous experiments. A more detailed description of these hypotheses will be provided in the introductory part of each experiment.

Experiment 1

In Experiment 1, the occurrence of a SNARC-like effect for tempo across a full temporal range was investigated, extending Experiment 1 by De Tommaso & Prpic (2020). In the original study, the authors failed to detect an effect and hypothesized that this was due to the large gaps between the stimuli (40 bpm, 80 bpm, 120 bpm, 160 bpm, 200 bpm). Accordingly, since the stimuli differed so drastically, we hypothesized that participants might have been unable to create a continuous representation of the temporal sequence. Our hypothesis was mostly speculative and was based mainly on the perceptual features of tempo (Gestalt proximity principle) rather than on the SNARC literature. Compared to the previous study, in the present experiment the gap between tempos was reduced, by testing a larger number of stimuli (40 bpm, 56 bpm, 72 bpm, 88 bpm, 104 bpm, 120 bpm, 136 bpm, 152 bpm, 168 bpm, 184 bpm, 200 bpm). Therefore, the aim of Experiment 1 is to investigate the occurrence of a SNARC-like effect for tempo, in a full range of bpm (40–200 bpm) with short intervals (16 bpm) between the stimuli.

Method

Participants

Thirty university students (M = 5, F = 25; Mage = 20.80, SD = 5.08) participated to experiment. To determine the sample size, G*Power software was used, with the following parameters for a two-tailed one-sample t- test on the mean of regression slopes: a power of 80%, a significance level (α) of 0.05 and an expected effect size (Cohen’s d) of 0.56, based on De Tommaso & Prpic (2020; Experiment 3). The calculated required sample size for the study was found to be 28 participants. All Participants were psychology students from University of Trieste and received academic credits for their participation. All of them reported that their reading/writing direction was left-to-right that and that they had normal hearing and normal or corrected to normal vision. None of them reported being a musician. Moreover, all of them reported that their psychophysiological state was not affected by alcohol consumption or insufficient sleep in the last 24 h (Murgia et al., 2020). The experiment was approved by the University of Trieste Ethics Committee and conducted with the ethical standards established by the Declaration of Helsinki and. Before data collection, each participant provided written informed consent.

Apparatus and stimuli

The experiment was programmed in the Psychopy software version 3.0 (Peirce, 2007). It was performed on Intel Core i5 10th generation pc (RAM: 8 Gb; Windows 10 operation system). Participants’ responses were collected using the first (most left) key and the fifth (most right) key of a response box with five keys. Headphones (Sony MDR-ZX110NA) with noise cancelling were employed to provide participants with stimuli; the volume was constant and set at a comfortable level for all the participants.

Eleven audio files composed by sequences of metronome beats with full range tempo were used as stimuli (40 bpm, 56 bpm, 72 bpm, 88 bpm, 104 bpm, 120 bpm, 136 bpm, 152 bpm, 168 bpm, 184 bpm, 200 bpm). In detail, the 120 bpm sequences served as reference stimulus while the other sequences were used as target stimuli. The duration of these audio sequences was 3,000 ms.

Procedure

All the participants were examined separately in a quiet laboratory under the supervision of the experimenter who maintained experimental conditions. Participants were asked to sit comfortably in front of the middle of the screen with the response box that was arranged in the middle line of their body. Participants were then required to minimise the movement and locate their left index finger on the first key of the response box (i.e., the leftmost key) and their right index finger on the fifth key of the response box (i.e., the rightmost key). Both the importance of speed and accuracy were stressed.

The task consisted in judging whether tempo of the target sequences was slower or faster than the reference sequence, by pressing one of the two response keys as quickly and accurately as possible, according to the instructions. In detail, the experiment was composed of two experimental blocks: a congruent block and an incongruent block. The condition in which the left key (vs. the right key) was a correct response to the slow tempo (vs. the fast tempo) was a congruent block. Conversely, the condition in which the right key (vs. the left key) was a correct response to the slow tempo (vs. the fast tempo) was an incongruent block. The experiment was made in two versions (A, B): one started with the congruent block followed by the incongruent block, while the other started with the incongruent block followed by the congruent one. The number of participants assigned to the two versions was equal and the order of presentation of the two blocks was counterbalanced among participants.

The experiment started with an instruction which explained the task and indicated the correct response keys. Additional explanations were provided by the experimenter verbally. Then, the participants were involved in practice block during which they got feedback if their responses were correct or not. Feedback appeared immediately after they had answered and was presented for 1,500 ms on the screen. The experimenter was present during the first practice in the research lab to make sure that the participants understood the procedures. Finally, after familiarized themselves with the task, the participants started the experimental block where no feedback were provided.

In both practice and experimental blocks, a fixation cross was presented for 1,500 ms at the beginning of each trial, followed by the reference sequence that was presented for 3,000 ms and the word “Reference” that was on the screen along the sound. Then, a fixation cross (between the reference and the target sequence) was presented with a random duration of 700 or 1,000 ms, followed by the target sequence, which was presented for 3,000 ms along with the word “Target”. Participants could respond as soon as they were presented with the target stimulus.

In the practice sessions, before each experimental block, participants performed 10 trials (one for each tempo, in random order), while in the experimental sessions they performed 60 trials for each block (six for each tempo, in random order).

This procedure was the same used in the original Experiment 1 by De Tommaso & Prpic (2020). However, there were a few differences: we increased the number of stimuli (10 vs. 4), consequently reducing the gap between tempos (16 bpm vs. 40 bpm), and we tested more participants (30 vs. 18).

Data analysis

The independent variables were Tempo (ten levels: 40 bpm, 56 bpm, 72 bpm, 88 bpm, 104 bpm, 136 bpm, 152 bpm, 168 bpm, 184 bpm, 200 bpm), and Hand (two levels: left and right). The dependent variable was the response times (RTs).

A total of two participants (6.66%) were removed from the study due to the high number of errors (i.e., number of incorrect responses over the threshold of 20%). They were removed since they provided the 45% and 50% of incorrect responses, respectively. We considered also to exclude any participants with over 50% of missing values under each specific level of our variables (e.g., left responses when target was slower than reference); no other participants were removed due this exclusion criterion. Moreover, the incorrect responses (4.85%) and outliers response times (1.75%) were excluded from the analyses (response times were considered as outlier if they were shorter than 120 ms or longer than the average response times of each participant plus 3 standard deviations, based on the same criteria used by Ishihara et al. (2008)).

The statistical software Jamovi 2.3.21 was used for data analysis. First, we calculated the differences in response times (dRTs) between the right and left hands for the Tempo variable (dRT = RT (right hand) - RT (left hand)). Positive dRT values indicated faster left-hand responses, while negative values indicated faster right-hand responses. Then, we conducted a linear regression using dRTs as the dependent variable to calculate individual regression slopes for each participant; negative values for regression slopes suggested a negative slope (SNARC-like effect). Finally, to evaluate whether the mean of regression slopes significantly deviated from zero, a one-sample t-test was performed.

Moreover, we analysed the absolute response times using a repeated-measures analysis of variance (ANOVA) with a 2x10 design, considering as factors Hand (two levels: left and right) and Tempo (ten levels: 40 bpm, 56 bpm, 72 bpm, 88 bpm, 104 bpm, 136 bpm, 152 bpm, 168 bpm, 184 bpm, 200 bpm).

Results and discussion

The one-sample t- test revealed that the mean of regression slopes significantly deviated from zero (t(27) = −3.29; p = . 003; d = −.622). The ANOVA revealed a significant main effect for Tempo (F(9, 243) = 74.60; p < .001; ηp2= .734) and a main effect for hand approaching significance (F(1, 27) = 3.72; p = .064; ηp2= .121). Moreover, a significant interaction between Tempo and Hand emerged (F(9, 243) = 3.95; p < .001; ηp2 = .128). The main effect for tempo revealed that stimuli in the fast range were responded faster. This was probably because information needed to provide a response was available earlier than for stimuli in the slow range, which required a longer listening time before a response decision could be made. The marginally significant main effect for hand revealed that responses were overall faster with the right hand, suggesting a response advantage typical of right-handed participants. Finally, the interaction between tempo and hand shows that slow tempos were generally responded faster with the left hand and fast tempos with the right hand, suggesting the presence of a SNARC-like effect for tempo (see Fig. 1). The results of post-hoc tests are reported in the OSF page of the project (link: https://osf.io/purzt/?view_only=e5bdc0b3f7e44eed97a115afa0475bb8).

Figure 1 (A) shows the mean of absolute response time (RTs) of the left and right hands for the full range tempo (slow: 40 bpm, 56 bpm, 72 bpm, 88 bpm, 104 bpm; fast: 136 bpm, 152 bpm, 168 bpm, 184 bpm, 200 bpm). (B) shows the mean dRTs (right hand–left hand) for each tempo. Positive dRTs show faster left-hand responses; negative dRTS show faster right-hand responses.

Errors bars represent standard error of the mean.

Experiment 1 was designed to extend the Experiment 1 by De Tommaso & Prpic (2020) in which a full tempo range was used, with large intervals between stimuli’s bpm (40 bpm), and no effect emerged there. The results of the present experiment showed a significant SNARC-like effect, with faster left-hand responses to relatively slow tempos and faster right-hand responses to relatively fast tempos. This effect was confirmed by both the one-sample t- test on B regression coefficients and the ANOVA. Our interpretation of this finding is that the large number of stimuli (10, with 16 bpm intervals) employed in this experiment elicited the representation of the temporal range along a continuum, facilitating the occurrence of spatial associations. Furthermore, results also showed an atypical pattern with absolute RTs remaining constant for slow tempos and decreasing rapidly for fast tempos (see Fig. 1; the discussion of this pattern will be provided in detail in the general discussion).

Experiment 2

In Experiment 2 we aimed to further investigate the occurrence of SNARC-like effects when separately testing the stimuli in the slow and fast tempo ranges, using a within-participants design. Indeed, in De Tommaso & Prpic (2020) the slow and fast tempo ranges (Experiments 2 and 3) were tested between participants and different outcomes occurred, namely a SNARC-like effect for the fast tempo range and no significant associations for the slow tempo range. It is possible that the different results in the two experiments were due to the interindividual variability of participants, revealing a sampling bias. Indeed, although there is no direct evidence of the role of individual differences on SNARC-like effects, recent studies suggest that the SNARC effect is only revealed in a subgroup of participants (i.e., fewer than 50% of participants; Cipora et al., 2024).

Accordingly, it is possible that in the study by De Tommaso & Prpic (2020), a larger number of participants predisposed to show a SNARC-like effect was sampled for the fast tempo condition, while a smaller number of participants was sampled for the slow tempo condition. This was even more likely due to the small samples tested. Therefore, assuming that a similar bias might occur also with SNARC-like effects, we conducted a single session within-participants experiment—in order to reduce the interindividual variability.

Method

Participants

Thirty students (M = 5, F = 25; Mage = 19.62, SD = 0.81) were recruited for this experiment. The sample size was calculated using the same parameters used in Experiment 1.

As in Experiment 1, all participants were psychology students recruited at University of Trieste who received academic credits for the participation. The sample had the same characteristics as the one of Experiment 1 (all participants were not musicians, had a left to right reading/writing direction, normal hearing, and no one reported alcohol consumption or insufficient sleep in the last 24 h). This experiment was conducted with the same ethical standards as Experiment 1.

Apparatus and stimuli

This experiment was performed online (due to the COVID 19 pandemic) with a generated code that was given to the participants to maintain anonymity. Participants were initially directed to the information sheet, which was hosted on Qualtrics (Qualtrics, Provo, UT, USA). Once they confirmed their willingness to proceed, they were redirected to experiment. The experiment was first programmed in the Psychopy software version 3.0 (Peirce, 2007) and later uploaded to Pavlovia which hosted it online. Participants could perform the experiment using their computer or laptop, with either a Windows or Mac operation system. Their responses were collected using the “A” and “L” keys (i.e., the left and right keys, respectively) on computer keyboards. Moreover, participants were asked to use headphones to perform the experiment, however, direct enforcement of this requirement was not possible.

Similar to Experiment 1, ten sequences of metronome beat at different tempos (40, 56, 72, 88, 104, 133, 150, 167, 184 and 201 bpm) were used as stimuli. In detail, five sequences (40, 56, 72, 88 and 104 bpm) were used for the slow tempo condition while the other five sequences (133, 150, 167, 184 and 201 bpm) were used for the fast tempo condition. The sequences at 72 bpm and 167 bpm served as reference stimulus in the slow and fast condition, respectively,

Procedure

As in Experiment 1, participants were required to judge whether tempo of the target sequences was slower or faster than the reference sequence, by pressing the left key (i.e., “A”) or the right key (i.e., “L”), according to the instructions. The presentation and the duration of the stimuli were the same as in Experiment 1.

This experiment was composed of four experimental blocks: two were used for the slow tempo conditions and the other two were used for the fast tempo conditions. Specifically, both slow and fast conditions had a congruent and incongruent block. In the two congruent blocks participants were required to respond with left key (“A”) if the target was slower than reference and with the right key (“L”) if the target was faster than the reference. In the two incongruent blocks the response assignment was reversed (i.e., participants had to respond with the left key (“A”) if the target was faster than the reference and with the right key (“L”) if the target was slower than the reference). To avoid order biases, the experiment was conducted in four versions (A, B, C, D). The order of the two conditions was counterbalanced (50% of the participants started with fast tempos and 50% with slow tempos), as well as the sequence of congruent/incongruent blocks (congruent-incongruent-congruent-incongruent vs. incongruent-congruent-incongruent-congruent). Specifically, in the A version of the experiment participants started with the fast tempo condition and the incongruent block; in the B version they started with the fast tempo condition and the congruent block; in the C version they started with the slow tempo condition and the incongruent block; in the D version they started with the slow tempo condition and the congruent block. Participants were equally distributed among the four versions of the experiment.

As in Experiment 1, participants always performed a practice block before the experimental block. In each practice block participants performed 12 trials (48 in total) while in each experimental block they performed 40 trials (160 in total).

Data analysis

Data analyses were similar to Experiment 1. The only difference was that separately analyses of t-test for the slow and fast conditions were conducted. Moreover, a paired sample t-test to compare the mean of the regression slopes of the two conditions was conducted. Finally, we conducted a 2 × 2 × 4 ANOVA, considering as factors Condition (slow and fast range), Hand (left and right) and the level of the Tempo conditions (“1,2,3,4” that were 40 bpm, 56 bpm, 88 bpm, 104 bpm for the slow tempo range, and 133 bpm, 150 bpm, 184 bpm, 201 bpm for the fast range).

Overall, incorrect responses (9.19%), outliers (1.44%) were removed based on the exclusion criteria used in Experiment 1.

Results and discussion

In the slow tempo condition, the analysis of the one-sample t- test did not reveal a significant deviation from zero in the mean of regression slopes (t(29) = −1.44; p = .161; d = −.263). In the fast tempo condition, the one-sample t- test revealed a significant deviation from zero in the mean of regression slopes (t(29) = −2.16; p = .039; d = −.395). Finally, the paired sample t-test did not reveal a significant difference in the means of regression slopes between slow and fast conditions (t(29) = .246; p = .807; d = −.044). The ANOVA revealed significant main effects for Tempo (F(3, 87) = 87.70; p < .001; ηp2= .751) and Condition (F(1, 29) = 97.01; p < .001; ηp2= .770), while no main effect was observed for Hand (F(1, 29) = .003; p > .05; ηp2= .000). Moreover, the analyses revealed a significant interactions between Tempo and Hand (F(3, 87) = 6.13; p < .001; ηp2= .174) and between Condition and Tempo (F(3, 87) = 7.24; p < .001; ηp2= .200) while the Condition x Hand x Tempo interaction was not significant (F(3, 87) = .040; p > .05; ηp2= .001). The main effects for Tempo and for Condition, and the interaction between Condition and Tempo suggest that different listening times for tempos influenced the response times pattern (these results will be discussed in detail in the general discussion). Importantly, the interaction between Tempo and Hand revealed a SNARC-like effect for tempo, while the not significant interaction Condition x Hand x Tempo suggested that this effect did not vary between conditions. The results of post-hoc tests for this experiment are reported in the OSF page of the project (link: https://osf.io/purzt/?view_only=e5bdc0b3f7e44eed97a115afa0475bb8).

Additionally, to examine whether the order of conditions influenced the results, the t-tests on regression slopes were conducted again using only data collected on the first condition for each participant. The results substantially confirmed what was observed in the main analyses. Specifically, in the slow tempo condition, no significant effect was found (t(13) = −0.313; p = .759; d = −.083), while in the fast tempo condition (t(15) = −2.72; p = .016; d = −.681) a significant effect emerged. Finally, since the percentage of incorrect responses were relatively high (9.19%), analyses on accuracy were performed. To do this, two paired-samples t-test were conducted to compare the average number of incorrect responses for congruent and incongruent blocks, in both slow and fast tempo conditions. The results revealed that the average number of incorrect responses did not vary between congruent and incongruent blocks; this was consistently observed both in the slow tempo condition (t(24) = .382; p = .706; d = .076) and in the fast tempo condition (t(27) = .706; p = .486; d = .133).

The results of Experiment 2 showed a SNARC-like effect, with faster left-hand responses to relatively slow tempos and faster right-hand responses to relatively fast tempos, in the fast range condition. This result is consistent with previous literature (De Tommaso & Prpic, 2020; Mariconda et al., 2024). As for the slow tempo range, the results are difficult to interpret. On the one hand, the one sample t- test does not show a significant effect, on the other hand the paired sample t-test and the ANOVA suggests that there is no difference between the slow and fast range conditions, with the ANOVA interaction showing an overall SNARC-like effect. Altogether, it is plausible that the effect in the slow tempo range condition–if any–is weak.

General Discussion

In the present study, we extended the experiments by De Tommaso & Prpic (2020). Specifically, we aimed to test the robustness of the SNARC-like effect for tempo and to extend previous results, addressing some open questions/methodological issues emerged from previous studies. To do so, in Experiment 1 (full tempo range) we extended the original Experiment 1 of De Tommaso & Prpic (2020) by reducing the gaps between tempos and thus testing a larger number of stimuli. In Experiment 2, to further investigate the spatial associations with slow tempos, we replicate the original Experiment 2 (slow tempo range) and Experiment 3 (fast range) by adopting a within-participants design. Overall, our results revealed the SNARC-like effect in the full range of tempos (Experiment 1) and confirmed the effect in the fast tempo range (Experiment 2—fast), while in the slow tempo range results are still ambiguous (Experiment 2).

Experiment 1 was an extension of Experiment 1 by De Tommaso & Prpic (2020), in which participants were required to listen to a reference sequence of beats and to judge whether the following target sequence was faster or slower than the reference. Our speculative hypothesis (based on Gestalt’s principle) was that, in the original study, the large gaps between the stimuli (four stimuli with 40 bpm intervals) might have determined the absence of perceptual unity, which in turn could have led to the absence of a SNARC-like effect. Consequently, in the present experiment, the perceptual unity of the temporal sequences was strengthened in accordance with the Gestalt’s proximity principle (stimuli tend to be aggregated into perceptual units if they are presented close to each other; Todorovic, 2008), by increasing the number of stimuli (10 vs. 4 as in the original study) and consequently reducing the gap between tempos (16 bpm vs. 40 bpm). Results revealed faster left-hand responses for relatively slow tempos and faster right-hand responses for relatively fast tempos, suggesting that the SNARC-like effect for tempo can be extended across a full tempo range. Therefore, it is plausible that the absence of effect found in the original study was actually due to the dramatic differences between stimuli, which might have hindered participants from forming a continuous representation of the temporal sequence.

Experiment 2 was a replication—in a within-participants design—of Experiment 2 (slow tempo range) and Experiment 3 (fast tempo range) by De Tommaso & Prpic (2020). In the original study, a SNARC-like effect was only revealed in the fast tempo range, while no effect was found in the slow tempo range; in the present experiment, the occurrence of the SNARC-like effect for the fast tempo range was clearly replicated, while in the slow tempo range condition contradictory results emerged. On the one hand, the one sample t-test did not reveal a SNARC-like effect in slow tempo range; on the other hand, neither the paired sample t- test, nor the ANOVA revealed different patterns for slow and fast tempo ranges. It is difficult to interpret such results, due to the absence of clear proofs in either direction.

The absent (or weak) SNARC-like effect in the slow tempo range is surprising and have already created a debate in the ‘SNARC’ community (see Wood, Shaki & Fischer, 2021; Mariconda et al., 2024), also because all SNARC and SNARC-like effects typically occur across different (numerical or non-numerical) magnitude ranges. Indeed, these effects seem to be related to the relative magnitude rather than the absolute one (although recent findings suggest that also absolute magnitude might play a role; Roth et al., 2023. For instance, the role of relative magnitude was clearly shown in the study by Dehaene, Bossini & Giraux (1993; Experiment 3) in which the central numbers (4 and 5) could be either associated to the right or to the left depending on the range of stimuli (0 to 5 or 4 to 9, respectively). Therefore, it is surprising that in the study by De Tommaso & Prpic (2020) no SNARC-like effect in the slow tempo range was found, and that in the present study clear evidence is still missing.

A potential explanation for the lack of clear evidence of a SNARC-like effect in the slow tempo range regards the relatively large gaps between the stimuli. Indeed, in percentage terms, a gap of 16 bpm is much larger in the slow tempo range, compared to the fast tempo range. For instance, from 40 bpm to 56 bpm there is a 40% increase, while from 136 bpm to 152 bpm there is a less than 12% increase. Thus, in the slow tempo range the gaps between stimuli might still be too large to create a perceptual unity of the temporal sequence, consequently hindering the occurrence of the SNARC-like effect. This explanation is consistent with the conclusion of our Experiment 1, in which the reduction of the gaps between the stimuli (compared to the original study by De Tommaso & Prpic (2020)) was considered as a crucial factor for the occurrence of the SNARC-like effect in the full tempo range. Moreover, we cannot exclude that the absence of perceptual unity could also influence the perceptual unity of each individual stimulus, that is, in the slowest tempos each beat might be perceived as a distinct sound rather than as a part of a sequence. However, it is plausible that a SNARC-like effect can occur also in the slow tempo range with the stimuli employed in the present study; indeed, they probably elicit a small effect which could be detected only testing a larger number of participants. In our opinion, to strengthen the effect (and to make it detectable with a sample size similar to that of the present study), it would be necessary to increase the number of the stimuli within the same range, consequently reducing the gaps between the stimuli and enhancing the perceptual unity of the temporal sequence.

Altogether, our study provides evidence that is in line with the ATOM (“A Theory Of Magnitude”) model proposed by Walsh (2003). Specifically, the spatial mapping of music tempo is in line with the prediction that every magnitude should elicit a Spatial Quantity Association of Response Codes (SQUARC) effect. Indeed, differently from the previous study by De Tommaso & Prpic (2020), our results revealed a SNARC-like effect across a full tempo range. Consequently, this result suggests that music tempo might be spatially represented like other magnitudes. However, clear evidence of a SNARC-like effect within the slow tempo range is still missing, thus suggesting that the range of temporal stimuli can play a role; it cannot be excluded that also neural differences in processing slow and fast tempos (see a recent work by Thibault, Albouy & Grondin, 2023) might affect spatial associations for this specific category of magnitude. That said, it is noteworthy that, although consistent evidence for SNARC-like effects exists in different magnitude domains, a few ATOM-driven failed attempts are reported in the literature (Cleland et al., 2020; Hohol, Szymanek & Cipora, 2024).

Additionally, another notable result regards the absolute response times found in the two tempo ranges. Notably, these are significantly faster in the fast tempo range (approximately 950 ms) compared to the slow tempo range (approximately 1250 ms), in line with previous literature (De Tommaso & Prpic, 2020; Mariconda et al., 2024). Based on the model proposed by Gevers et al. (2006), the magnitude of the SNARC effect should increase with slower response times, therefore a more pronounced SNARC-like effect should be expected in the slow tempo range rather than in the fast range. However, the results of the present study show a clear SNARC-like effect only for the fast range, apparently in contrast with the model by Gevers and colleagues (Gevers et al., 2006). This apparent contradiction can be explained by examining both the stimuli and the experimental paradigm. In the slow tempo range (in particular for extremely slow tempos, e.g., 40 and 56), participants needed more time to respond, as there was a longer duration between consecutive beats; in other words, they had to spend more time to gain enough information to complete the task, resulting in longer RTs. Conversely, in the fast range condition (in particular for extremely fast tempos, e.g., 184 and 201 bpm), the shorter intervals between beats could allow participants to process information earlier and, consequently, respond faster. Therefore, the longer response times in the slow tempo condition are probably not due to a longer processing of information, but to a longer wait to gain enough information to respond.

Interestingly, an atypical pattern was revealed in Experiment 1 (see Fig. 1), with absolute RTs remaining nearly flat for slow tempos and decreasing rapidly for fast tempos in the full tempo range. This pattern is probably due to both the longer listening time described previously and the presence of a distance effect. For instance, 40 bpm should theoretically be judged quickly because it is far from the reference (i.e., 120 bpm), however participants responded slowly due to the long listening time. Conversely, although 104 bpm required a shorter listening time, participants responded slowly because this tempo was close to the reference. Therefore, the pattern observed with slower tempos might partially mask the distance effect. In contrast, at faster tempos, absolute response times decreased as the tempo moved away from the reference, in line with the distance effect. This pattern was probably strengthened by the listening time which decreased with increasing bpm. For instance, the response times were extremely fast for bpm far from the reference (i.e., 184 and 201 bpm), both due to the distance effect and the shorter listening time. A similar pattern can be found also in Experiment 2 (see Figs. 2 and 3); in the slow tempo range the longer listening time partially masks the distance effect, while in the fast tempo range absolute response times are more in line with the typical pattern of the distance effect due to shorter listening time.

Figure 2 (A) shows the mean of absolute response time (Rts) of the left and right hands for slow range condition (40 bpm, 56 bpm, 88 bpm, 104 bpm). (B) shows the mean of absolute response time (Rts) of the left and right hands fast range condition (133 bpm, 150 bpm, 184 bpm, 201 bpm).

Errors bars represent standard error of the mean.

Figure 3 (A) shows the mean dRTs (right hand–left hand) for slow range condition (40 bpm, 56 bpm, 88 bpm, 104 bpm). (B) shows the mean dRTs (right hand–left hand) for fast range condition (133 bpm, 150 bpm, 184 bpm, 201 bpm).

Positive dRTs show faster left-hand responses; negative dRTS show faster right-hand responses. Errors bars represent standard error of the mean.

Furthermore, a proof indicating the occurrence of a distance effect in this study derives from the comparison of the absolute response times for the slowest tempos between Experiment 1 and Experiment 2. In Experiment 1 the reference (120 bpm) was far from the slowest target tempos (i.e., 40, 56 and 88 bpm), while in Experiment 2 the reference (72 bpm) was close to them; the variation of the reference (keeping constant the targets and, consequently, the listening time) produces an increase of the absolute response times by over 300 ms (i.e., RTs for 40, 56 and 88 bpm were approximately equal to 1,000 ms in Experiment 1 vs. 1,300 ms in Experiment 2). The same reasoning can be applied to the fastest tempos. Overall, these observations further confirm that the absolute response times in our study were actually influenced by both the listening time and the distance effect.

The strength of the present study is that it provides a clear replication of a previously studied effect, which is the occurrence of a SNARC-like effect in the fast tempo range; moreover, it extends this effect to the full tempo range. In contrast, current evidence does not allow to generalize these results to the slow tempo range, leaving still open the debate on the different spatial associations for slow and fast tempo ranges (De Tommaso & Prpic, 2020; Mariconda et al., 2024; Wood, Shaki & Fischer, 2021). Although this study is the first one suggesting that a SNARC-like effect is plausible even in the slow tempo range, clear evidence of this effect is not provided and further research is needed. Follow-up studies should replicate our slow range condition, either with a larger number of participants or with a reduced gap between the stimuli, which should be comparable—in percentage terms—with the gaps of the fast range condition.

Conclusion

The present study was an extension of the study by De Tommaso & Prpic (2020) and was conducted with a twofold aim: (1) to extend the SNARC-like effect for tempo in a full range of bpm (40–200 bpm); (2) to further investigate the occurrence of spatial associations in the slow and fast tempo ranges (40–104 bpm and 133–201 bpm, respectively). Experiment 1 indicates that when the perceptual unity of the temporal sequence is strengthened in the full tempo range (i.e., by reducing the gaps between the stimuli), the SNARC-like effect occurs. Experiment 2 further confirms the occurrence of the SNARC-like effect in the fast tempo range, whereas it provides controversial results in the slow tempo range. The lack of a clear outcome of spatial associations in the slow tempo range needs to be further explored. On the one hand, it is plausible that a SNARC-like effect might be elicited in this range as well, but previous experiments failed to detect this (probably weak) effect because they were not properly designed; on the other hand, it cannot be excluded that slow and fast tempo are differently processed, consequently revealing different patterns of spatial associations.

Supplemental Information

Supplemental Information 1 Raw data - Experiment 1

Response times and correct/incorrect responses for all conditions described in the paper.

Supplemental Information 2 File Readme - Experiment 1

Supplemental Information 3 Raw data - Experiment 2

Response times and correct/incorrect responses for all conditions described in the paper.

Supplemental Information 4 File Readme - Experiment 2

Additional Information and Declarations

Competing Interests

Author Contributions

Human Ethics

Data Availability

Valter Prpic is an Academic Editor for PeerJ.

Alberto Mariconda conceived and designed the experiments, performed the experiments, analyzed the data, prepared figures and/or tables, authored or reviewed drafts of the article, and approved the final draft.

Mauro Murgia conceived and designed the experiments, authored or reviewed drafts of the article, and approved the final draft.

Matteo De Tommaso conceived and designed the experiments, authored or reviewed drafts of the article, and approved the final draft.

Tiziano Agostini conceived and designed the experiments, authored or reviewed drafts of the article, and approved the final draft.

Valter Prpic conceived and designed the experiments, authored or reviewed drafts of the article, and approved the final draft.

The following information was supplied relating to ethical approvals (i.e., approving body and any reference numbers):

The University of Trieste Ethics Committee granted Ethical approval to carry out the study within its facilities

The following information was supplied regarding data availability:

The raw data of Experiments 1 and 2 are available in the Supplemental Files.

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
