# Peer review of "SNARC-like effect for tempo is consistent for fast and full tempo ranges but still controversial for slow tempo range"

_PeerJ, doi:10.7717/peerj.18009_

## Round 0.1 · original submission · Major Revisions

As you can see, both reviewers found your work to be sound and valuable, and I agree with them. They also offered several practical suggestions to improve this work, none of which are critical. These suggestions are precise and to the point, so I will not reiterate them. In revising your work, please consider all the comments you’ve received and provide a detailed response letter.

In addition to the reviewers’ comments, I would be pleased to read some additional remarks explaining why the data for Experiment 2 were collected online rather than in the laboratory using the same apparatus as in Experiment 1. Lab-based and online experiments can certainly lead to similar results, but in the case of auditory stimuli, transitioning to online experiments might be more challenging than for purely visual experiments.

·

Basic reporting

The manuscript is well-organized and clearly written, addressing a significant research inquiry with suitable methodologies. The discussion of relevant literature is comprehensive, and the figures are exhaustive and clear. The primary findings are appropriately examined and substantiated by empirical evidence. However, there are certain aspects pertaining to hypotheses, experimental procedures, and results that in my opinion warrant a more thorough examination and discussion.

Experimental design

Experiment 1 is portrayed as an attempt to conceptually replicate Experiment 1 in De Tommaso and Prpic (2020). However, the appropriateness of the term "conceptual replication" in this context is uncertain. The authors' hypothesis suggests that by increasing the "density" of stimuli within the range of 40-200 bpm compared to the original study, they aim to alter participants' representations of the stimuli. This conceptual departure from (not replication of) the original study aims to assess whether a change in stimulus representation results in a consistent SNARC-like effect. Given this rationale, the term "extension", which is sometimes used already in the manuscript, appears to be a more fitting description than "conceptual replication." If the authors concur with this perspective, it is advisable to replace the term "conceptual replication" throughout the manuscript.

On lines 120-126, a concise overview of the main hypotheses is presented. To ensure that readers do not perceive these hypotheses as superficially defined, it would be beneficial to add a sentence indicating that a more detailed description of the hypotheses will follow in subsequent sections of the manuscript.

Lines 133-134. This hypothesis appears intuitive, yet a more in-depth technical discussion regarding its rationale is warranted. It is important to provide justification for positing that a representation of stimuli along a continuum is more conducive to eliciting a SNARC-like effect compared to a more categorical representation. Are there existing data in the literature that support this hypothesis, or is it primarily based on the authors' intuitions? If the latter, it is advisable to explicitly label the hypothesis as speculative.

Line 166. I think that it is worth specifying that the duration of these audio sequences was 3000 ms.

Line 186. It may be worth specifying whether the number of participants assigned to the two versions of the experiment was equal or unequal. Additionally, please verify the correctness of the term “randomly assigned.” Do the authors actually mean that the order of presentation of the two blocks was counterbalanced among participants?

Lines 195-201. Based on the results of Experiment 1, particularly the smaller absolute RTs recorded for faster tempos compared to slower tempos, it appears that participants could respond as soon as they were presented with the target stimulus. However, this crucial information is currently missing in the procedure description.

End of Procedure section. Since Experiment 1 is presented as an attempt to extend De Tommaso and Prpic’s (2020) study, it may be worth adding a couple of sentences focusing on the comparison between the methods of the two studies (sample size, procedure, etc.) .

Line 201. Please clarify whether you mean 60 trials in total (30 per block) or 60 trials per block. Additionally, it is important to specify if there was only one practice session at the beginning of the experiment, or one practice session before each experimental block.

Lines 206-207. Please clarify what you mean by “high number of errors” and provide a justification for the removal of the data from these two participants.

Lines 228-230. The ANOVA results are presented without any further commentary on the main effect of Tempo or the interaction. Ideally, the authors should report the results of post-hoc tests to explore the origins of the statistically significant results, or at least provide a qualitative description of the meaning of the effects. As currently presented, these ANOVA results are not very informative. I would also recommend that the authors report the results for the non-statistically significant effects, for the sake of replicability and possible meta-analytic exploration.

Figure 1. The pattern of results displayed in panel a is quite atypical for this type of study, with absolute RTs remaining fairly constant for slow tempos and decreasing rapidly for fast tempos. Since the reader has to wait until the General Discussion for an explanation, I recommend that the authors either anticipate the discussion of the pattern of absolute RTs or add a sentence indicating that a discussion of this interesting result will be provided in the General Discussion.

Lines 255-262. The rationale for Experiment 2 could be clarified to enhance its coherence. While I concur with the authors that the inconsistent results reported by De Tommaso and Prpic (2020) in a between-subject design might stem from inter-individual variability, the connection between this point and Cipora et al.'s (2019) finding—that only about 50% of participants exhibit a consistent SNARC effect—is not entirely clear. This connection seems relevant only if there's an assumption that in De Tommaso and Prpic (2020), a sampling bias occurred, where many participants predisposed to showing a SNARC-like effect were sampled for the fast tempo condition, while fewer such participants were sampled for the slow tempo condition. Without explicit reference to a possible sampling bias, referencing Cipora et al.'s (2019) work might be confusing. Furthermore, the lack of reliability of the SNARC effect (Roth et al., 2023) may be irrelevant here, as participants are tested only once.

Lines 301-306. This passage is confusing, partly due to issues with the grammatical structure of sentences. Please revise it to ensure that the different orders of conditions are clearly described.

Lines 326-329. Same as in Experiment 1: solely reporting the statistically significant main and interaction effects without conducting post-hoc tests or providing further commentary on the meaning of these effects is, in my opinion, not very informative.

Validity of the findings

Building upon the results of a previous study, the findings from the two experiments presented here contribute to providing a more comprehensive and precise understanding of the processes underlying the spatial representation of tempo, particularly regarding the distinct representations of fast and slow tempos. Therefore, I believe that the present work aligns well with the editorial scopes of PeerJ.

About the data, I would recommend including a brief "read me" document to clarify the meaning of the data contained in each column of both files, as in some cases, it may not be self-evident.

Lines 387-390. As highlighted in one of my previous comments, it is important to inform the reader about the source of this hypothesis: does it find any support in the literature, or is it purely speculative?

Is it plausible that the presence or absence of perceptual unity could influence not only the global representation of the entire set of presented stimuli as forming or not forming a perceptual continuum, but also the perceptual unity of each individual stimulus presented? It's conceivable that, unlike stimuli in the fast range, stimuli in the slow range were perceived as distinct sounds rather than as a unified sequence. This perception might have hindered a clear perceptual representation of the stimulus as "slower" or "faster" than the reference, potentially resulting in a null or diminished SNARC-like effect. While the results indicate that participants were able to perform the comparison task with stimuli in the slow range, it cannot be excluded that the process underlying responses to stimuli in the slow range was of a more high-level nature compared to that underlying responses to stimuli in the fast range, which could be more perceptual in nature. I’m aware that this is pure speculation, however I would appreciate a comment on this hypothesis, either in the manuscript or in the authors’ responses.

Line 463. I would add a couple of sentences to clarify that the specific nature of the stimuli, particularly the direct relationship between tempo and the time needed to accumulate sufficient information before providing a response, precludes drawing any reliable conclusions about the distance effect. In other words, the effect described by the authors might mask the distance effect.

Additional comments

Some typos that need to be corrected:
Line 112. this -> that
Line 135. gaps -> gap
Line 152. no one -> all of them
Line 214. for were -> was
Line 268. has -> had
Line 271. standard -> standards
Lines 290 and 293. As Experiment 1 -> As in Experiment 1
Lines 294-295. for slow tempo conditions -> for the slow tempo condition
Line 321. find -> reveal (or show)

Reviewer 2 ·

Basic reporting

Several studies have found that time is cognitively represented in a spatial way, i.e., as the mental timeline, similarily to number, which is represented along the mental number line. In the domain of time, the spatial representation is indexed by the presence of effects resembling the SNARC effect (Spatial-Numerical Association of Response Codes). Mariconda et al.’s study constitutes an attempt to replicate conceptually the former study by De Tommaso and Prpic where the SNARC-like effect has been found for fast tempo range. In two studies described in the manuscript, the authors wanted to investigate the phenomenon of interest in the full tempo range (Exp1) and in the slow and fast tempos (Exp2). In a nutshell, in Exp1, they found the effect for the full range. In Exp2 they found the effect for fast range but not to slow range.

I am very pleased to have the opportunity to review this manuscript, and I believe that the results presented there enlarge our understanding of associations between space and other magnitudes.

The study is generally well-designed, the description of the procedure is sufficiently precise to allow replication, and the data are shared according to open science policies. The manuscript is generally well-written, however, I have some comments, the incorporation of which would, in my opinion, make the manuscript more accessible to a wider audience of psychological/cognitive scientists.

Abstract:
-L21: I suggest avoiding unexplained abbreviations; here, I would recommend explaining, or alternatively removing, “SNARC-like effect for tempo.”
Introduction:
-Authors start with (spatial-temporal associations, next embed them in the ATOM framework, and then go to spatial-musical associations. This is not a strong suggestion, but perhaps it would be better to start with ATOM, and then go to temporal and musical magnitudes.
-L50-1: “Evidence suggests that the spatial alignment of this mental timeline is influenced by embodied and cultural factors.” It is not clear what the authors mean by embodied and cultural factors - the authors list only writing/reading direction. In the context of the original SNARC effect, writing/reading direction is considered to be a cultural one, wherein possible cultural factors include finger-counting habits (Fischer 2008, Cortex; Hohol, Wołoszyn, Cipora, 2022, Acta Psychologica). Could the author specify any embodied factor for the SNARC-like effect for tempo?
L60: “(; Torralbo et al., 2006….) - typo or missing reference.
L72-3: [SNARC] “…which demonstrates that digits are spatially represented in the human mind.” I personally agree with the interpretation of the SNARC effect in the context of the ATOM context, but I think it would be fair to introduce, at least a reference, an alternative interpretation of the original SNARC.
L74: “non-symbolic numerals” - please consider replacing numerals with numerosities. L74-82: Authors list SNARC-like effects for several kinds of magnitude. This is good, but I think it would be fair to list also ATOM-driven failed attempts to reveal SNARC-like effect (Hohol, Szymanek, Cipora, 2024, Scientific Reports; Cleland, Corsico, White, Bull, 2020, Q. J. Exp. Psychol.).

Experiment 1
L129-30: “In Experiment 1, the occurrence of a SNARC-like effect for tempo across a full temporal range was investigated in a conceptual replication of Experiment 1 by De Tommaso and Prpic (2020).” Rather “as” (”as a conceptual replication”).
Apparatus and stimuli
L162-3: “the volume was constant and set at a comfortable level for participants.” - please specify whether a level was the same for all the participants.
Stimuli (the same applies to Exp2): the authors would consider sharing (as a furher suppl.material) with audio files used as stimuli.
Analysis
The analysis was performed expertly, and I have no remarks to this part.
Results
L230 - please remove 0 in partial eta sq to report this stat consistently.
L233 - “and no effect emerged.” => “and no effect emerged there.”

Experiment 2
I have no remarks to the description of Exp2.

General discussion
L411-412 - “…SNARC and SNARC-like effect typically occur…” - see my remark to the introduction; I think that listing null results of studies on SNARC-like effect would be good in this context.
L412 - I think “magnitudes” would be a better word than “ranges.”.
L430-1 “However, we cannot exclude that a SNARC-like effect can occur also in the slow tempo range with the stimuli employed in the present study” - once again, I think sharing your stimuli with the community would be nice.
One more comment at the end. You mentioned the issue of individual differences in the manuscript. Maybe, as in other SNARC-like musical effects, the musical experience of participants matters in revealing the phenomenon in the case of slow tempo.

Experimental design

The study is a conceptual replication, so it undertakes a well-defined research question. The description of the procedure is sufficiently precise to allow further replication.

Design - I should acknowledge that the sample is unbalanced in terms of gender, but to my knowledge, gender does not affect spatial representations of the magnitude investigated by the authors (and other magnitudes as well), so I do not think so this is a serious problem.

Validity of the findings

The analyses are conducted expertly. Conclusions are driven by the obtained results.

---

## Round 0.2 · accepted · Accept

Both the Reviewers and I are pleased with the revised version of the manuscript. I am happy to accept this work for publication. Congratulations!

·

Basic reporting

I thank the authors for having carefully addressed all of my concerns. I think that the manuscript can be accepted for publication in the current form.

Experimental design

\

Validity of the findings

\

Additional comments

\

Reviewer 2 ·

Basic reporting

Thank you for addressing all my comments, introducing the amendments, and sharing materials. I have no further substantial comments, thus I'm happy to recommend approval of the manuscript.

There is only one thing that should be addressed at the stage of proofreading. I have found small linguistic issues. Authors should correct them and double-check whether further (overlooked by me) typos have crept into the text during the revision.

lines 58, 407, 536, 543 0-missing dots after vs (should be vs., not vs)
line 600: should be "exists".

Best wishes, Mateusz Hohol

Experimental design

I have no further comments.

Validity of the findings

I have no further comments.